# Effect of EGCG Extracted from Green Tea against Largemouth Bass Virus Infection

**DOI:** 10.3390/v15010151

**Published:** 2023-01-03

**Authors:** Yuan Cheng, Mingzhu Liu, Qing Yu, Shuaishuai Huang, Shuyu Han, Jingu Shi, Hongling Wei, Jianwei Zou, Pengfei Li

**Affiliations:** 1Guangxi Key Laboratory of Aquatic Biotechnology and Modern Ecological Aquaculture, Guangxi Engineering Research Center for Fishery Major Diseases Control and Efficient Healthy Breeding Industrial Technology (GERCFT), Guangxi Academy of Sciences, Nanning 530000, China; 2China-ASEAN Modern Fishery Industry Technology Transfer Demonstration Center, Nanning 530000, China; 3Guangxi Fisheries Technology Extension Station, Nanning 530000, China; 4Beihai Fisheries Technology Extension Station, Beihai 536001, China

**Keywords:** antiviral activity, antiviral mechanism, aptamer, EGCG, largemouth bass virus

## Abstract

(1) Background: Largemouth bass virus (LMBV) is a major viral pathogen in largemouth bass (*Micropterus salmoides*) aquaculture that often causes high mortality and heavy economic losses, thus developing treatments to combat this pathogen is of great commercial importance. Green tea is a well-known medicinal plant that contains active ingredients with antiviral, antibacterial, and other biological activities. The goals of this study were to explore the effect and mechanism of green tea source compounds on LMBV and provide data to serve as the basis for the screening of targeted drugs in the future. In this study, we evaluated the effects of the main component of green tea, epigallocatechin-3-gallate (EGCG), against LMBV infection. (2) Methods: The safe working concentration of EGCG was identified by cell viability detection and light microscopy. The antiviral activity and mechanism of action of EGCG against LMBV infection were evaluated with light microscopy, an aptamer 6-carboxy-fluorescein-based fluorescent molecular probe, and reverse transcription quantitative PCR. (3) Results: The safe working concentration of EGCG was ≤10 μg/mL. EGCG showed significant anti-LMBV infection activity in a concentration-dependent manner, and it also destroyed the structure of virus particles. EGCG impacted the binding of virus particles to cell receptors and virus invasion into the host cells. Inhibitory effects of EGCG on LMBV particles, LMBV binding to the host-cell membrane, and LMBV invasion were 84.89%, 98.99%, and 95.23%, respectively. Meanwhile, the effects of EGCG subsequently were verified in vivo. The fatality rate of the LMBV + EGCG group was significantly lower than that of the LMBV group. (4) Conclusions: Our results suggest that EGCG has effective antiviral properties against LMBV and may be a candidate for the effective treatment and control of LMBV infections in largemouth bass aquaculture.

## 1. Introduction

Largemouth bass (*Micropterus salmoides*) is widely cultured around the world and is a significant and economically important aquaculture species [1]. Advancements in the regulation of reproductive growth and artificial breeding technologies have gradually transformed largemouth bass farming in China into an intensive aquaculture industry [2], and the aquaculture yield reached 619,519 tons in China in 2020 [3]. However, the rapid development of largemouth bass farming has been accompanied by frequent outbreaks of various pathogens. For example, largemouth sea bass virus (LMBV), which is a nucleoplasmic large DNA virus of the genus *Ranavirus* (family Iridoviridae), has caused significant economic losses and seriously impacted the global largemouth bass breeding industry [4,5]. At present, an effective control method for LMBV disease in largemouth bass has not been reported [6]. 

Therefore, effective antiviral drugs to combat LMBV infection are urgently needed. Yi et al. developed a DNA vaccine that can be injected through the base of the pectoral fin, and it conferred significant immune protection against iridovirus in largemouth bass [7]. However, the scope of vaccine application was limited because of its well-known defect: its failure to kill the virus and treat infected fish. Therefore, more effective treatment modalities need to be developed, and medicinal plants are promising candidates.

Medicinal plants contain a variety of active ingredients. Compared with chemical drugs, some medicinal plants have advantages such as lower toxicity, fewer side effects, and less resistance to pathogens [8,9]. The antiviral activity of polyphenols, the main ingredient of green tea, is well known [10]. Some studies have shown that tea polyphenols (TP) can inhibit the release of influenza virus from host cells by (1) competing with the second binding site of the sialic acid receptor on the surface of host cells to bind neuraminidase, thereby inhibiting the replication and spread of the virus [11,12], or (2) inhibiting influenza A virus endonuclease activity by binding to the active region of the RNA polymerase of the virus, thereby inhibiting RNA synthesis [13]. Catechins are the main components of TP, and they include (-) epigallocatechin gallate (EGCG), (-) epicatechin gallate (ECG), (-) epigallocatechin (EGC), and (-) epicatechin (EC) [14]. Among them, EGCG accounts for 50–60% of TP [15]. The phenolic hydroxyl groups on the B and C rings of EGCG have proton supply activity and considerable reducing capacity [16]. In addition, EGCG has many other active sites, with a variety of health functions and pharmacological effects, such as antibacterial, antiviral, anti-cancer and cancer prevention, lower blood lipid, and anti-allergy [17]. In one study, researchers found that EGCG inhibits grass carp reovirus infection in a *Ctenopharyngodon idellus* kidney cell line [18]. Yamaguchi et al. showed that EGCG had a destructive effect on viral particles [19]. Numerous researchers have reported that EGCG has a significant antiviral effect on the adsorption, invasion, replication, and release of different viruses [20,21,22,23,24,25,26,27,28]. 

In this study, we investigated the antiviral effects of EGCG against LMBV infection in order to provide a valuable reference for the development of safe and effective drugs against LMBV. We determined the safe and effective concentration of EGCG using light microscopy and CCK-8 enzyme-linked immunodetection. We also analyzed the antiviral activity and mechanism of action of EGCG in vitro using reverse transcription quantitative real-time polymerase chain reaction (RT-qPCR) and an aptamer 6-carboxy-fluorescein (FAM)-based fluorescent molecular probe.

## 2. Materials and Methods

### 2.1. Cells, Virus, and Aptamer

Fathead minnow (FHM) cells were used in this study, and LMBV (isolated from diseased largemouth bass) was used to infect the cultured cells. LMBV-infected fish cells are specifically recognized by the aptamer FAM-LBVA1, which was previously shown to be an excellent molecular probe for detecting LMBV infection [29]. LBVA1 was labeled with FAM and synthesized by Sangon Biotech (Shanghai, China). The EGCG used in this study was isolated from green tea (>98% purity) and purchased from Macklin Co., Ltd. (Shanghai, China). The EGCG was dissolved in dimethyl sulfoxide (DMSO; Sigma, Germany) and stored at −20 °C. The stock solution was diluted to the working concentration in L15 medium (pH 7.5). Healthy largemouth bass (6.0 ± 0.5 cm body length) were purchased from a fish farm in Nanning City, Guangxi, China. The fish were acclimated for 7 days before the experiment began.

### 2.2. Cell Viability Assay to Identify the Safe Concentration of EGCG

FHM cells (1.5 × 10^6^) were seeded in 96-well plates (Corning, Shanghai, China) at 28 °C for 24 h, and then different concentrations of EGCG (1.25, 2.5, 5, 10, and 20 μg/mL) were added to wells; cells in the control group were treated with equal amounts of Leibovitz’s L15 medium. Cells were observed for signs of cytotoxicity using light microscopy after 48 h of incubation at 28 °C. Next, 10 μL of CCK-8 solution (Beyotime, Shanghai, China) was added to the wells, and cells were incubated for 4 h at 28 °C. The absorbance at 450 nm was measured using an ELISA plate reader (Thermo Fisher Scientific, Waltham, MA, USA). The cell viability rate was calculated according to the following equation: Cell viability rate = (X − control 2)/(control 1 − control 2) × 100%, where X is the absorbance of cells treated with aqueous extracts at different concentrations; control 1 is the absorbance results of untreated cells; and control 2 is the absorbance of cells treated with L15. 

### 2.3. Fluorescence Microscopy Observation of the Cytoskeleton at the Safe Concentration of EGCG

The cellular effect of EGCG at its safe working concentration was evaluated by fluorescence microscopy [30]. FHM cells (1.5 × 10^6^) were cultured in 35 mm glass-bottom dishes (Cellvis, Burlington, ON, Canada) at 28 °C for 48 h and then incubated with EGCG (10 μg/mL) for 48 h at 28 °C. After incubation, the cells were washed with phosphate-buffered saline (PBS; pH 7.4) three times and subsequently dyed with fluorescein isothiocyanate-labeled anti-cytokeratin antibody. Fluorescence was observed by laser scanning confocal microscopy (LSCM, Nikon, C2, Tokyo, Japan). Untreated FHM cells served as the control group.

### 2.4. Gene Expression Measurement by RT-qPCR

FHM cells (1.5 × 10^6^) were cultured in 12-well plates at 28 °C for 18 h. After the respective treatments, the cells and supernatant in each well were collected to extract total RNA with TRIzol Reagent (Cwbio, Nanning, China). Total RNA was then reverse-transcribed into cDNA using a ReverTra Ace^®^ qPCR RT Kit (Toyobo, Osaka, Japan). LMBV infection was confirmed by the detection of immediate early protein (ICP46) transcripts. The expression of the target gene was normalized against that of β-actin. Table 1 shows the primers used for RT-qPCR. 

### 2.5. FAM-LBVA1 Monitoring of LMBV Infection

Aptamers are artificial chemical antibodies, and they have the advantages of strong specificity, high affinity and stability, and easy chemical synthesis [31]. We previously developed a molecular probe aptamer, FAM-LBVA1, which is highly accurate and efficient against LMBV [29]. In the current study, FHM cells (1.5 × 10^6^) were cultured in 12-well plates for 18 h at 28 °C. After the respective treatments, we used the FAM-labeled aptamer LBVA1 to identify LMBV infection by flow cytometry (Beckman Moflo XDP, Krefeld, Germany).

### 2.6. Cy5-LMBV Monitoring of LMBV Infection

LMBV virus particles were fluorescently labeled with Cy5 dye as previously described [32]. Virus particles obtained by high-speed centrifugation were incubated with Cy5 dye for 2 h at room temperature with gentle vortexing. The unbound dye was removed and washed by three high-speed centrifugations at 25,000 g at 4 °C for 1 h. The virions bound to the dye were collected by high-speed centrifugation at 25,000 g for 1 h at 4 °C, and then they were washed three times with TN buffer. Finally, Cy5-labeled LMBV (Cy5-LMBV) pellets were suspended in TN buffer and passed through 0.22 μm pore size filters and examined by transmission electron microscopy.

### 2.7. Analysis of the Antiviral Effect of EGCG against LMBV In Vitro

FHM cells (1.5 × 10^6^) were seeded in 12-well plates at 28 °C and incubated for 18 h. High, medium, and low (10, 5, 2.5 μg/mL) safe working concentrations of EGCG, respectively, were premixed with LMBV (multiplicity of infection (MOI) = 1) in 800 μL of L15 medium and added to the FHM cells. In pilot studies, we found that ICP46 gene expression between the LMBV group and LMBV-DMSO group was no significant difference, this demonstrates that DMSO did not have an anti-LMBV effect, and the data are available in the Appendix A. Therefore, FHM cells supplemented with L15 medium alone served as control group 1, and FHM cells incubated in L15 medium containing only LMBV served as control group 2. At 48 h post-infection, cytopathic effects (CPEs) caused by LMBV infection were visualized by light microscopy. Cells in the wells (including the culture medium) were used to extract total RNA, and the anti-LMBV activity of EGCG was assessed by RT-qPCR. 

### 2.8. Analysis of the Anti-LMBV Mechanism of EGCG: Effects on Virus Particle Structure

FHM cells (1.5 × 10^6^) were seeded in 12-well plates and incubated for 18 h. The cells were then incubated with the safe concentration of EGCG (containing LMBV, MOI = 1) at 4 °C for 2 h. The mixture was centrifuged at 4 °C at 25,000 g for 30 min, and the supernatant was discarded. Viral particles clustered at the bottom of the cells were dissolved in TN buffer, added to the cells of 12-well plates, and incubated for 48 h. FHM cells in medium without the addition of EGCG and LMBV served as control group 1, and LMBV-infected FHM cells (without EGCG) served as control group 2. After 48 h of culture, the CPEs were observed by light microscopy. The ICP46 expression in cells and culture supernatants were analyzed by RT-qPCR.

### 2.9. Analysis of the Anti-LMBV Mechanism by Which EGCG Affects the Virus Infection Process

FHM cells (1.5 × 10^6^) were incubated in 12-well plates for 18 h. Control group 1 was cultured with L15 medium, and control group 2 was cultured with L15 medium containing LMBV (MOI = 1) alone at 4 °C. After 30 min, the supernatant was discarded and the FHM cells in each group were washed with L15 medium. Cells were kept in L15 medium at 28 °C for 12 h. Finally, total RNA was extracted for RT-qPCR analysis.

Test 1: To study the effect of EGCG on the adsorption stage of LMBV infection, we mixed EGCG and LMBV (MOI = 1) with L15 medium and added the mixture to the FHM cells. After incubation for 30 min at 4 °C, the supernatant medium was discarded, the cells were washed gently twice with L15 medium, and finally, the cells were cultured with new L15 medium at 28 °C for 12 h. Cells were then harvested for RT-qPCR analysis.

We also analyzed the role of EGCG on the adsorption phase of LMBV using Cy5-LMBV. Healthy FHM cells that had been incubated for 18 h in 12-well plates were incubated with Cy5-tagged LMBV (MOI = 1) and EGCG for 2 h at 4 °C. The supernatant and cells with adherent growth at the bottom of the wells were collected, washed twice with PBS, and resuspended in PBS for flow cytometric analysis. Control groups were cells treated with L15 and cells treated with Cy5-LMBV only.

Test 2: To investigate the effect of EGCG on virus invasion into cells, we added LMBV (MOI = 1) to FHM cells alone at 4 °C for 30 min. After discarding the supernatant medium and gently washing the cells twice, EGCG was added and the cells were cultured at 28 °C for 2 h. The supernatant was removed, cells were washed twice with medium, new L15 medium was added, and the cells were cultured at 28 °C. Cells were collected 10 h later for RT-qPCR analysis.

Flow cytometry was also used to analyze the effect of EGCG on LMBV invasion into host cells. FHM cells were cultured with Cy5-tagged LMBV (MOI = 1) only for 2 h at 4 °C. The supernatant and cells with adherent growth at the bottom of the wells were collected and washed twice with PBS. EGCG was added and the mixture was incubated for 2 h at 28 °C. Control groups were cells treated with L15 and cells treated with Cy5-LMBV only. Before flow cytometry, each group was treated with proteinase K for 2 min at room temperature to remove LMBV particles that had not yet entered the cells.

Test 3: To explore the effect of EGCG on viral replication in the cell, LMBV (MOI = 1) was added to FHM cells, and the mixture was incubated at 4 °C for 30 min. After washing twice with L15 medium, the cells were incubated at 28 °C for 2 h. Next, the cells and EGCG were co-incubated for another 8 h, followed by incubation in L15 for 2 h. The expression of viral ICP46 was measured by RT-qPCR.

### 2.10. Quantification of Inhibitory Effects of EGCG against Absorption, Invasion, and Replication of LMBV

The inhibitory percentage of EGCG components on LMBV infection was evaluated by RT-qPCR. The following equation was used to calculate the inhibition rate: Inhibitory rate = 1 − (X − control)/(test − control) × 100%, where X is the RT-qPCR detection result of each group treated with EGCG and LMBV; test it the RT-qPCR result of infection with LMBV alone; and control is the RT-qPCR result of cells without treatment.

### 2.11. Antiviral Activity of EGCG against LMBV Infection In Vivo

The 360 largemouth bass were divided into each of the following groups in triplicate: (i) DMSO-treated negative control; (ii) LMBV only; and (iii) LMBV and green tea component. Each largemouth bass was injected intraperitoneally with a 50 μL treatment volume. Total RNA was extracted from the spleen of fish at 24, 48, and 72 h, and five fish were taken from each group at different times, respectively. Mortality was recorded daily for 10 days. The results were calculated as the means ± SD.

### 2.12. Statistical Analysis

The average value of three independent experiments was calculated. Intergroup differences were compared by one-way analysis of variance using SPSS statistical software (IBM, Armonk, NY, USA). For all analyses, *p* < 0.01 was considered to indicate statistically significant differences.

## 3. Results

### 3.1. Safe Concentration of EGCG

After FHM cells were treated for 48 h with different concentrations of EGCG, the cells were analyzed for cytotoxic effects. The cell state was first observed by light microscopy. FHM cells maintained normal growth when cultured with EGCG at the safe working concentration (10 μg/mL) or lower, with no significant changes in cell morphology versus control cells (Figure 1A). In contrast, FHM cells treated with a higher concentration of EGCG (20 μg/mL) showed significant morphological lesions. The main visible changes were detachment from the culture plate, rounding, and aggregation into clusters. We then checked the viability of the cells in each group using CCK-8, and the results were consistent with those observed by light microscopy (Figure 1B).

The cytoskeleton is a major mechanical structure of the cell and plays an important role in cellular function (Fletcher and Marlins, 2010). The cytoskeleton was visualized using fluorescein isothiocyanate-labeled anti-cytokeratin antibodies (Figure 1C). Compared with the control cells, the cytoskeleton of cells treated with EGCG (10 μg/mL) did not collapse or fracture, indicating that no significant cytotoxic effect occurred at the safe concentration of EGCG. 

### 3.2. Antiviral Activity of EGCG

We first estimated the antiviral activity of EGCG against LMBV infection using light microscopy, FAM-LBVA1 detection, and RT-qPCR analysis. Compared to large numbers of typical CPEs appearing in the LMBV-infected cells, few CPEs appeared in cells incubated with LMBV and high, medium, and low safe concentrations of EGCG (Figure 2A). FAM-LBVA1 flow cytometry analysis showed that the fluorescence strength of FHM cells incubated with both LMBV and EGCG at all safe working concentrations was much lower than that of cells incubated with LMBV alone (Figure 2B). Furthermore, at 48 h post-culture, the ICP46 expression decreased significantly in the EGCG-treated group, and the trend was consistent with the flow cytometry results (Figure 2C). These results indicated that EGCG exhibited anti-LMBV activity in a dose-dependent manner, with 10 μg/mL EGCG having the best antiviral activity.

### 3.3. Anti-LMBV Mechanism of EGCG: EGCG Destroys Virus Particles

Light microscopy showed that only a few CPEs developed in the FHM cells infected with the purified virus pretreated with EGCG (Figure 3A). RT-qPCR showed that the ICP46 expression level was significantly lower in the cells in the LMBV group pretreated with EGCG than in the group incubated with LMBV particles (Figure 3B). Together, these results confirmed that EGCG destroyed virus particles before they entered the cell and reduced virus expression in the host cells.

### 3.4. Anti-LMBV Mechanism of EGCG: EGCG Affects the LMBV Infection Process

RT-qPCR (Figure 4) and flow cytometry (Figure 5) revealed the effects of EGCG on the different stages of LMBV infection in FHM cells. The expression level of ICP46 in the experimental group was lowest in the adsorption stage, followed by the invasion stage. In the replication phase, however, ICP46 expression did not differ significantly between the EGCG pre-treatment LMBV group and the virus-only group. This result indicated that EGCG interfered with LMBV binding to targets on the host cells (Figure 4). The Cy5 fluorescence strength in cells in the EGCG treatment group was much lower than that in cells in the LMBV-infected group (Figure 5), confirming that both adsorption and invasion phases of LMBV infection were inhibited to varying degrees by EGCG.

### 3.5. Inhibitory Effects of EGCG on Different Phases of LMBV Infection

The RT-qPCR data for ICP46 gene expression throughout the infection stage were used to assess the inhibitory effect of EGCG at all stages of LMBV infection. The inhibitory effects of EGCG on LMBV particles, LMBV binding to host cells, LMBV invasion, and LMBV replication in cells were 84.89%, 98.99%, 95.23%, and −11.38%, respectively (Figure 6). Thus, EGCG has significant antiviral activity against LMBV infection and mainly acts on LMBV ontology, contact between the virus and the host cell membrane, and the process of virus entry into the cell.

### 3.6. Antiviral Activity of EGCG against LMBV Infection In Vivo

Anti-LMBV activity of EGCG was detected in vivo. Largemouth bass were injected intraperitoneally with LMBV mixed with EGCG at its safe working concentration of 10 μg/mL (LMBV + EGCG group). The control groups were injected intraperitoneally with diluted DMSO (con) and LMBV, respectively. Liver, spleen, and kidney tissues collected at 24, 48, and 72 h were used for antiviral analysis. The expression of ICP46 in the LMBV + EGCG group was significantly reduced at 24, 48, and 72 h post-infection (hpi) compared with the group injected with LMBV alone (Figure 7). The fatality rate of largemouth bass infected with LMBV reached 100% at 8 days (Figure 8). The cumulative mortality of the con group and LMBV + EGCG group was 3.33% and 15% at 10 days, respectively. The survival rate of the largemouth bass co-injected with EGCG and LMBV was significantly higher than that of the largemouth bass injected with only LMBV.

## 4. Discussion

Green tea is an important medicinal plant. The effects of its major constituent polyphenols on body health are well known and include antiviral, antioxidant, and antimicrobial activities [33,34]. Catechins are active compounds that have been isolated and identified from green tea [35]. The most abundant catechins, such as EGCG and ECG, have excellent antiviral effects against influenza virus and human immunodeficiency virus (HIV) [36,37], and EGCG has effects against hepatitis virus, herpes viruses, and Singapore grouper iridovirus (SGIV) [9,38,39]. We previously explored the antiviral effects of some green tea extracts against SGIV and found that EGCG could combat SGIV infection with high inhibition (98.31%) [9]. However, the role of EGCG against LMBV had not been explored prior to this study. Therefore, we investigated the antiviral activities and mechanisms of action of EGCG to gain valuable data that can be used to develop effective antiviral therapeutics against LMBV.

Fish cell lines can be used in place of experimental animals and are widely used for toxicity assessment in vitro [40]. The cytoskeleton is an interconnected lattice that consists of different types of microtubules, actin, and intermediate filaments, and it plays important roles in various cellular functions, such as supporting cell shape and assisting in whole-cell movement and organelle movement. Significant cytoskeletal defects can occur in diseased cells, including altered axonal transport, microtubule stability, and actin dynamics [41,42,43]. Additionally, the cytoskeleton in the host cell plays an indispensable role during viral infection. For example, spring viremia associated with carp virus infection can induce the collapse of the cytoskeletal fiber system, ring structure, and filament depolymerization [43]. Wang et al. suggested that SGIV can be moved via pipelines in cells and that depolymerization of actin filaments or microtubules might significantly affect SGIV movement in cells [41]. In the current study, possible cytotoxic effects of EGCG were evaluated on FHM cells by fluorescence observations. The cytoskeleton of cells incubated with EGCG (10 μg/mL) remained normal, indicating that this safe working concentration of EGCG had no cytotoxic effects. The cell morphology and cell viability assays produced the same results.

During viral infection, viruses first adsorb to the host cell membrane and then enter the cell by cytosis. After entering the cell, viruses initiate the expression and replication of viral genes at specific sites [32,41,44]. Understanding how antiviral drugs exert their antiviral effects is essential for developing antiviral medicine. Therefore, systemic studies were performed to explore the anti-LMBV mechanisms of action of EGCG. Yamaguchi et al. reported that EGCG may bind to the HIV-1 viral envelope surface, deform the envelope phospholipids, and prompt virus disruption [19]. Undoubtedly, the intact viral structure is requisite for viral infection, so we first evaluated the effect of EGCG on LMBV particles and found that pre-incubation with EGCG reduced the subsequent viral expression in FHM cells. Thus, we propose that the pathogenicity of LMBV is indeed damaged by EGCG action on the virus particles.

Investigation of the effect of EGCG on different phases of the LMBV life cycle in FHM cells revealed that EGCG interfered with the binding of LMBV to the host cell targets (98.99%) and obstructed the invasion of host cells by LMBV (95.23%). These results show that EGCG had the best inhibitory effect during the stage of receptor binding to the host cell membrane, which is key to protecting cells against LMBV infection. This coincides with the best antiviral viability in the LMBV life cycle and is a key link in LMBV infection. Nakayama et al. previously reported that EGCG might agglutinate virions by binding to hemagglutinin and preventing the virus from attaching to the cell surface, thus inhibiting influenza A and B viruses from infecting MDCK cells [45]. Mitsuyo et al. found that EGCG interacted with bovine coronaviruses and hindered virus adsorption to MDBK cells [46]. Furthermore, Wang et al. reported that EGCG prevented grass carp reovirus adsorption and invasion of host cells by blocking the laminin receptors on the host cell surface [11]. In the current study, EGCG interfered with LMBV entry into the host cells (95.23%). 

It is well known that tea polyphenols or their monomeric components can impede HIV infection through different avenues. For example, catechins, theaflavins, and their derivatives were found to compete with gp120 of HIV for binding to CD4 on the cell surface [47] or by directly targeting gp41 of HIV to inhibit the formation of the virus six-helix bundle structure (6-HB), thereby blocking the virus from invading the host cells [48,49]. Carneiro et al. studied the anti-Zika virus (ZIKV) effects of EGCG using Vero E6 cells and found that EGCG significantly inhibited ZIKV invasion into the cells. Subsequently, Sharma et al. confirmed that EGCG disrupted the virion structure by binding to the viral envelope protein, thus blocking ZIKV invasion [50]. A recent report showed that EGCG inhibited the activity of the hepatitis B virus core promoter and that subsequent viral replication by ERK1/2-signaling mediated the downregulation of hepatocyte nuclear factor 4α [28]. However, we did not detect the inhibitory effects of EGCG on the replication stage in LMBV. Ciesek et al. suggested that hepatitis C virus (HCV) replication and assembly of full-length HCV genomes were unaffected by EGCG, and they illustrated that EGCG could not inhibit the essential HCV NS3/4A serine protease in an HCV replication setting [51]. This result is consistent with the lack of inhibitory effects of EGCG on the replication stage of LMBV in our study. This indicated that EGCG could not inhibit the activity of replication-related enzymes of LMBV in host cells.

The effects of EGCG subsequently were verified in vivo. The results were consistent with those of the in vitro study. The fatality rate of the LMBV + EGCG group was significantly lower than that of the LMBV group. These results suggest that green tea extract EGCG had a good antiviral effect on LMBV both in vivo and in vitro.

## 5. Conclusions

The green tea aqueous extract EGCG displayed excellent antiviral activities against LMBV infection, and it inhibited LMBV infection in a dose-dependent manner. Furthermore, significant antiviral effects were present during most stages of viral infection. These data indicate that EGCG extracted from green tea has direct and host-mediated antiviral effects against LMBV and therefore has great potential as a drug to control LMBV infection in aquaculture.

## Figures and Tables

**Figure 1 viruses-15-00151-f001:**
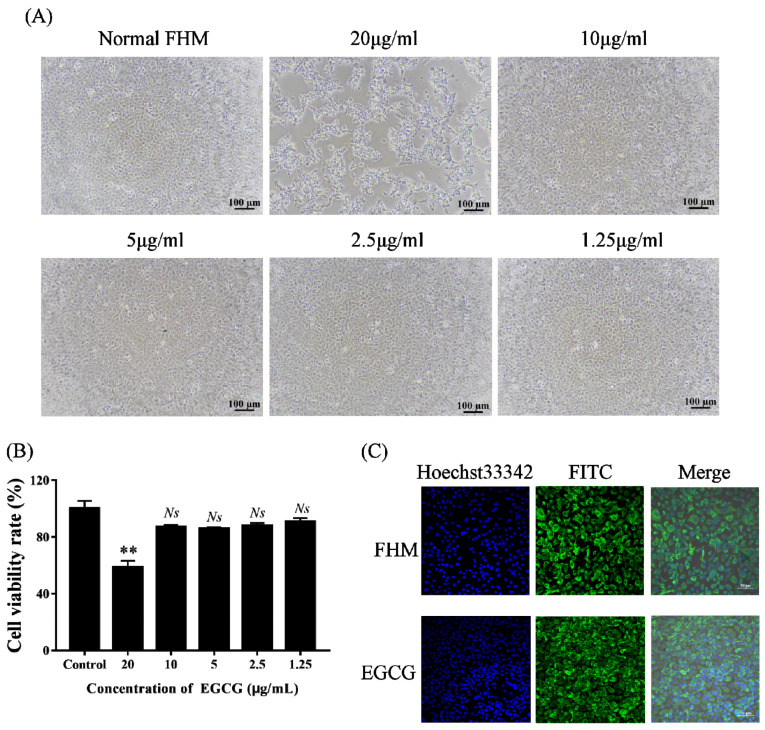
Safe concentration of EGCG. (**A**) The morphological changes of FHM cells incubated with different concentrations of EGCG were observed by light microscopy. Pathological morphological changes, including rounding, shrinking, and detachment from the culture plate, were not detected in FHM cells incubated with EGCG at concentrations of 10 μg/mL or lower. Scale bar indicates 100 μm. (**B**) CCK-8 was used to detect cell viability of EGCG-treated FHM cells. The viabilities of cells cultured with 1.25–10 μg/mL EGCG were the same as those of the control cells, illustrating that EGCG ≤ 10 μg/mL had no significant cytotoxic effects on FHM cell viability. (** *p* < 0.01; *NS*, no significant difference). The same below. (**C**) Fluorescein isothiocyanate-labeled anti-cytokeratin antibodies were applied to observe the cytoskeleton. Compared with cells in the control group, the cytoskeleton of cells incubated with EGCG (10 μg/mL) was kept normal. Scale bar: 50 μm.

**Figure 2 viruses-15-00151-f002:**
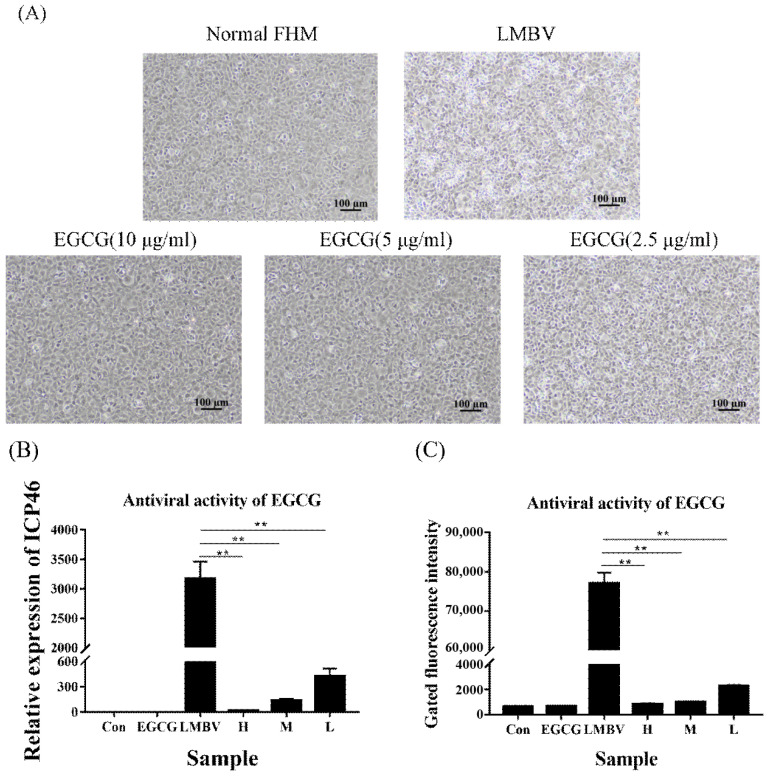
Analysis of the antiviral activity of EGCG. (**A**) Light microscopy of CPEs in each group; almost no CPEs appeared in the non-treatment group. Compared to the large number of CPEs in the virus infection group, CPEs in the EGCE treatment group were obviously lacking. Among the treatment groups, the number of CPEs was lowest at 10 μg/mL, followed by 5 μg/mL and then 2.5 μg/mL. (**B**) The results of flow cytometry analysis showed that the fluorescence intensity of the FHM cells incubated with different concentrations of EGCG as well as LMBV was significantly weaker than that of the cells treated with LMBV only. (**C**) RT-qPCR detection showed that the expression of viral ICP46 in FHM cells was significantly reduced after incubation with different concentrations of EGCG. All of these results indicated that EGCG inhibited LMBV infection in a dose-dependent manner. ** indicates *p* < 0.01.

**Figure 3 viruses-15-00151-f003:**
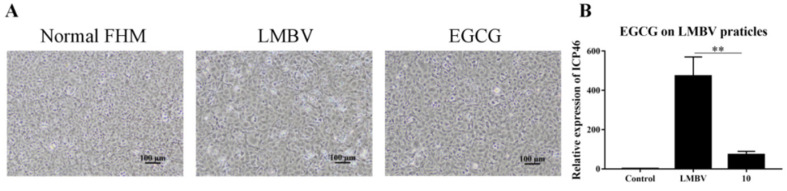
Action of EGCG against LMBV particles based on RT-qPCR analysis and light microscopy observations. (**A**) CPEs in FHM cells incubated with purified virus pretreated with EGCG were less abundant than in cells incubated with virus alone. (**B**) The expression level of ICP46 in the EGCG pretreatment group was significantly lower than that in the purified LMBV group without EGCG. ** indicates *p* < 0.01.

**Figure 4 viruses-15-00151-f004:**
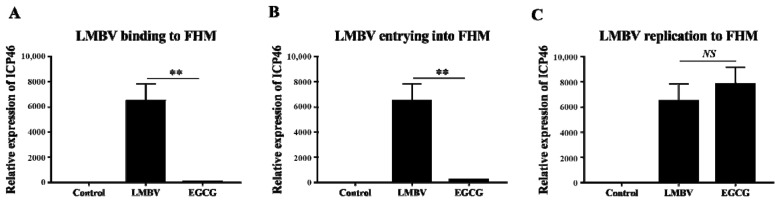
Antiviral ability of EGCG in the various stages of viral infection was analyzed by RT-qPCR. (**A**) When EGCG was added to the host cells at the adsorption stage of LMBV, the expression of ICP46 was significantly inhibited. (**B**) When EGCG was added to the host cells at the invasion period of LMBV, the expression of ICP46 was significantly inhibited. (**C**) When LMBV had already entered the host cell, the addition of EGCG did not show a significant inhibitory effect on the expression of ICP46. ** indicates *p* < 0.01, *NS* indicates no significant difference.

**Figure 5 viruses-15-00151-f005:**
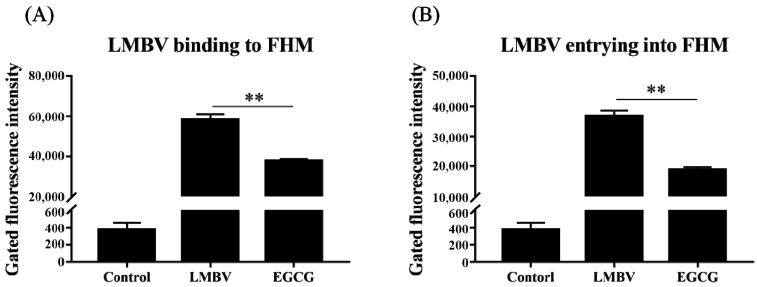
The inhibitory effects of EGCG on LMBV binding (**A**) and entry (**B**) into host cells were illustrated by flow cytometry analysis. Cy5-fluorescence signals were significantly decreased in EGCG-treated cells compared with the control group. ** indicates *p* < 0.01.

**Figure 6 viruses-15-00151-f006:**
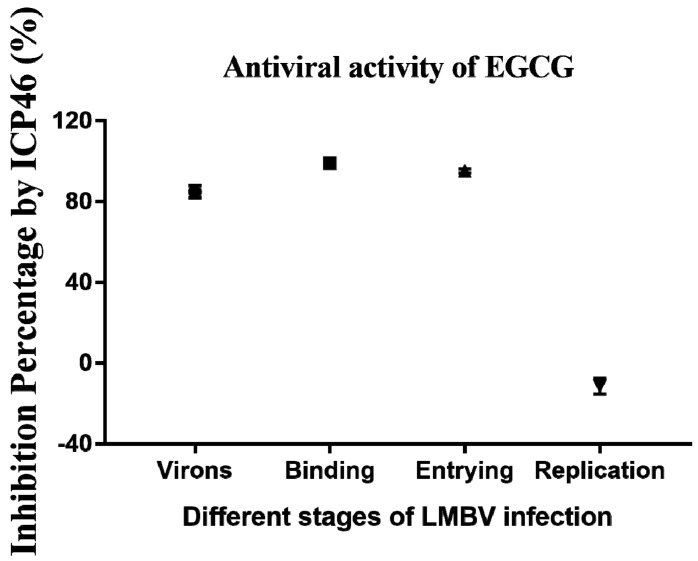
Inhibitory percentage of EGCG on different stages of LMBV infection. The inhibitory effects of EGCG on LMBV infection mainly acted on LMBV particles, the binding of LMBV to host cell membranes (test 1), and invasion of cells (test 2), and no inhibition was found in the replication phase (test 3). The inhibitory rates of virons, binding, entrying, replication stage were 84.89%, 98.99%, 95.23%, and −11.38%, respectively.

**Figure 7 viruses-15-00151-f007:**
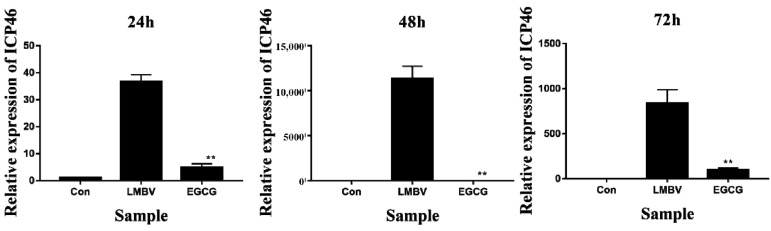
Antiviral activity of green tea component EGCG against LMBV in vivo. Largemouth bass were injected intraperitoneally with LMBV mixed with EGCG at its safe working concentration of 10 μg/mL. Spleen tissues were analyzed for antiviral activity at 24, 48, and 72 hpi. ** indicates *p* < 0.01.

**Figure 8 viruses-15-00151-f008:**
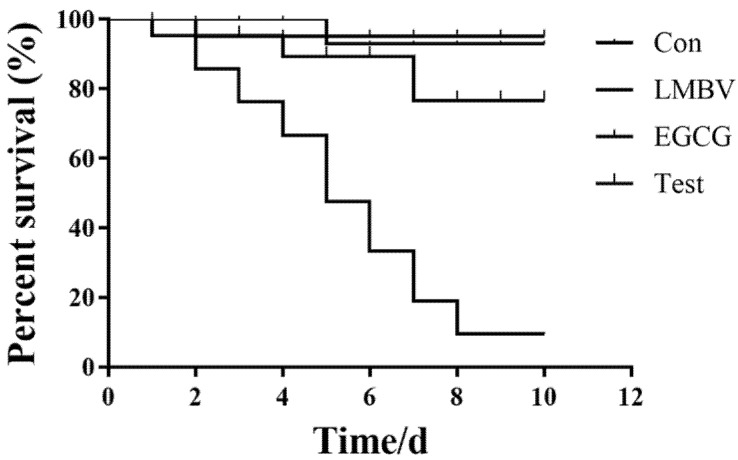
Cumulative mortality over 10 days. In the group of fish treated with LMBV only, the cumulative mortality reached 90% on day 8 post-infection. By contrast, in the test group, cumulative mortality was 20%. It indicated that EGCG had anti-LMBV activity in vivo.

**Table 1 viruses-15-00151-t001:** Primers used for detecting (LMBV) infection by RT-qPCR.

Primer	Sequences
qICP46-F	5′-CAACTGCAGACTGGTCCTGA-3′
qICP46-R	5′-AAAGCCTGTTGAGGAGACGA-3′
β-actin-F	5′-TCTTCCAGCCATCCTTCCTTGG-3′
β-actin-R	5′-CTGCATACGGTCAGCAATGCC-3′

## Data Availability

The data that support the findings of this study are available from the corresponding author upon reasonable request.

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
