# Peer review of "Effect of EGCG Extracted from Green Tea against Largemouth Bass Virus Infection"

_viruses, 2023, doi:10.3390/v15010151_

Round 1

Reviewer 1 Report (Previous Reviewer 1)

I have no more suggestion on the paper, as the authors have addressed all the revised comments. The paper could be published at its current status.

Author Response

Thanks for your recommendation

Reviewer 2 Report (New Reviewer)

In the manuscript “Effect of EGCG Extracted From Green Tea against Largemouth Bass Virus Infection”, Chen Yuan and colleagues evaluated the antiviral activity of EGCG against LMBV infection in vitro and in vivo, and explored the anti-LMBV mechanism of EGCG. The manuscript is informative for researchers and worthy of publication, but some problem need to be addressed as follow:

1. The in vivo experiment results are not mentioned in the abstract, the author should give the supplement.

2. The origin, size, health, and temporary rearing condition of largemouth bass should be described in detail in 2.1, but not in 2.11.

3. The authors need to determine that Fig.8 is indeed a cumulative mortality rate expressed as a percentage.

4. The authors should supply a note to Fig.8.

5. “In vitro” and “In vivo” in the last paragraph of discussion should be italicized.

6. Line 338, were groupers used in the test?

7. Figure 8 lacks a brief description. In figure 8, was SGIV used in the test? TPor EGCG?.

8. The format of reference should be correct to the same as the full paper.

9. In Results 3.6, The survival rate of the largemouth bass co-injected with EGCG and LMBV was significantly lower than that of the groupers injected with only LMBV. Check this description.

10. Authors used the immediate early protein ICP46 to evaluate the propagation of LMBV. Did authors have proof to identify this protein as an immediate early protein?

Author Response

Response to Reviewer 2 Comments

In the manuscript “Effect of EGCG Extracted from Green Tea against Largemouth Bass Virus Infection”, Chen Yuan and colleagues evaluated the antiviral activity of EGCG against LMBV infection in vitro and in vivo, and explored the anti-LMBV mechanism of EGCG. The manuscript is informative for researchers and worthy of publication, but some problem need to be addressed as follow:

  1. The in vivo experiment results are not mentioned in the abstract, the author should give the supplement.

Response: Thanks for your recommendation.

We had added it: Meanwhile, the effects of EGCG subsequently were verified in vivo. The fatality rate of the LMBV+ EGCG group was significantly lower than that of the LMBV group (Line 38-39).

  1. The origin, size, health, and temporary rearing condition of largemouth bass should be described in detail in 2.1, but not in 2.11.

Response: Thanks for your recommendation.

We have added it: Healthy largemouth bass (6.0 ± 0.5 cm body length), were purchased from a fish farm in Nanning City, Guangxi, China. The fish were acclimated for 7 days before the experiment (Line 99-101).

  1. The authors need to determine that Fig.8 is indeed a cumulative mortality rate expressed as a percentage.

 Response: Thanks for your recommendation. We applied percentages to exhibit the survival rates in Fig.8.

  1. The authors should supply a note to Fig.8.

Response: Thanks for your recommendation.

We have revised it in the highlighted manuscript: In the group of fish treated with LMBV only, the cumulative mortality reached 90% on day 8 post-infection. By contrast, the survival rate of test group was 80%. It indicated that EGCG had anti-LMBV activity in vivo (Line 354-356).

  1. “In vitro” and “In vivo” in the last paragraph of discussion should be italicized.

 Response: Thanks for your recommendation. We have revised them.

  1. Line 338, were groupers used in the test?

 Response: Thanks for your recommendation. We have corrected it (Line 343).

  1. Figure 8 lacks a brief description. In figure 8, was SGIV used in the test? “TP” or “EGCG”?

 Response: Thanks for your recommendation. We used largemouth bass and noted“The survival rate of the largemouth bass co-injected with EGCG and LMBV was significantly higher than that of the largemouth bass injected with only LMBV” in the article (Line 352).

  1. The format of reference should be correct to the same as the full paper.

Response: Thanks for your recommendation. We have revised them (line 561-565). 

1.Sharma, N.; Murali A.; Singh S K.; et al. ; Giri, R. Epigallocateching-allate, an active green tea compound inhibits the Zika virus entry into host cells via binding the envelope protein. International Journal of Biological Macromolecules, 2017, 104, 1046-1054.

2.Ciesek, S.; von, H. T.; Colpitts, C. C.; Schang, L. M.; Friesland, M.; Steinmann, J.; Manns, M. P.; Ott, M.; Wedemeyer, H.; Meuleman, P.; Pietschmann, T.; Steinmann, E. The green tea polyphenol, epigallocatechin-3-gallate, inhibits hepatitis C virus entry. Hepatology (Baltimore, Md.). 2011, 54(6): 1947-1955.

  1. In Results 3.6, the survival rate of the largemouth bass co-injected with EGCG and LMBV was significantly lower than that of the groupers injected with only LMBV. Check this description.

Response: Thanks for your recommendation. We have revised it. The survival rate of the largemouth bass co-injected with EGCG and LMBV was significantly higher than that of the largemouth bass injected with only LMBV (Line 342-343).

  1. Authors used the immediate early protein ICP46 to evaluate the propagation of LMBV. Did authors have proof to identify this protein as an immediate early protein?

Response: Thanks for your recommendation.

Liqun Xia had reported that “SGIV ORF162L encoding a putative homolog of ICP46 was identified and characterized. Interestingly, ICP46 could be found in all sequenced iridoviruses and is considered as a core gene of the family Iridoviridae. SGIV ICP46 was classified as an immediate-early (IE) gene during in vitro infection using drug inhibition analysis, reverse transcription polymerase chain reaction and Western blot analysis. Subcellular localization revealed that SGIV ICP46 was distributed predominantly in the cytoplasm. Furthermore, SGIV ICP46 proved to be a structural protein of the nucleocapsid; its overexpression could promote the growth of grouper embryonic cells and contribute to SGIV replication [1, 2].

Fig. The homology of ICP46 analyzed by the phylogenetic tree.

ICP46 was annotated as an immediate early gene in the LMBV genome from NCBI.  We compared this sequence with the ICP46 gene of other viruses and found that they clustered together.  Therefore, based on the results of ICP46 in SGIV, we speculated that the ICP46 used in this study was the immediate early protein of LMBV. 

[References]

[1] Xia L, Liang H, Huang Y, Ou-Yang Z, Qin Q. Identification and characterization of Singapore grouper iridovirus (SGIV) ORF162L, an immediate-early gene involved in cell growth control and viral replication. Virus Res. 2010 Jan; 147(1):30-9. doi: 10.1016/j.virusres.2009.09.015. Epub 2009 Oct 1. PMID: 19800375.

[2] Xia L Q, Chen J L, Zhang H L, et al. Identification of virion-associated transcriptional transactivator (VATT) of SGIV ICP46 promoter and their binding site on promoter [J]. Virology journal, 2019, 16(1): 1-14.

This manuscript is a resubmission of an earlier submission. The following is a list of the peer review reports and author responses from that submission.

Round 1

Reviewer 1 Report

I have no more suggestions for the paper, as the authors have addressed all comments from reviewers. The paper could be published at current status.

Reviewer 2 Report

In the present manuscript, authors demonstrated the effect of green tea component epigallocatechin-3-gallate (EGCG) against largemouth bass virus (LMBV) in vitro and in vivo as well as the mode of action of EGCG. Authors are requested to address the following concerns.

1. Could the authors calculate the CC50 and EC50? Only FHM cells were used in this study. Could the authors repeat cell viability and antiviral assay in other fish cells?

2. In Figure 7, the inhibitory effect of EGCG on different phases of LMBV infection was investigated. Specifically, largemouth bass were injected intraperitoneally with LMBV mixed with EGCG at its safe working concentration of 10 μg/mL (LMBV + EGCG group) (50 μL/ largemouth bass).

Why is the concentration of EGCG used here? That means 0.5μg EGCG/ largemouth bass?

Furthermore, how about EGCG treatment after LMBV infection instead of injection intraperitoneally with LMBV mixed with EGCG?

Which tissue does "sample" in Figure 7 represent? Please check.

What is the effect of EGCG on the mortality of LMBV-infected largemouth bass?

3. The overall writing is not good enough to publish and there are many errors. Please check the whole text.